# Giant charge-to-spin conversion in ferromagnet via spin-orbit coupling

Yuki Hibino [1✉], Tomohiro Taniguchi [1], Kay Yakushiji [1], Akio Fukushima[1], Hitoshi Kubota[1] & Shinji Yuasa [1]

Converting charge current into spin current via the spin Hall effect enables efficient manipulation of magnetization by electrical current. However, its geometrical restriction is a serious obstacle to device applications because it prevents switching of perpendicular magnetization in the absence of an external field. To resolve this issue, ferromagnetic materials have attracted attentions because their time reversal asymmetry induces magnetic-dependent charge-to-spin conversion that removes this restriction. Here, we achieved a large enhancement of magnetic-dependent charge-to-spin conversion by clarifying its mechanism. Through layer thickness dependence of the conversion efficiency, we revealed a coexistence of interfacial and bulk contributions to the magnetic-dependent charge-to-spin conversion. Moreover, the interfacial contribution to charge-to-spin conversion is found to be dominant and can be controlled via interfacial band engineering. The efficiency of charge-to-spin conversion in ferromagnet was found to be an order larger than that of other materials with reduced symmetry.

[1] National Institute of Advanced Industrial Science and Technology (AIST), Research Center for Emerging Computing Technologies, Tsukuba, Ibaraki 305-8568, Japan. ✉email: y-hibino@aist.go.jp

Charge-to-spin current conversion arising from spin–orbit coupling[1,2] is one of the most important phenomena in the field of spintronics. When current-driven spin current is injected into an adjacent ferromagnetic material (FM), spin–orbit torques (SOTs) act on the magnetic moment of the ferromagnet, leading to magnetization dynamics such as polarity switching[3–5]. Highly efficient spin-current generation has been achieved using non-magnetic materials (NMs), such as heavy metals (Pt, $\beta$-Ta, and $\beta$-W)[4–8] and topological insulators[9,10], where the spin Hall effect and spin-momentum locking at the surface state play a crucial role. However, it has also been revealed that SOTs generated from these materials cannot achieve switching in perpendicular magnetization without assistance from an external field, which is a major obstacle to developing high-density spintronics devices such as SOT-driven non-volatile memory. This issue arises from the geometrical restriction in SOTs. The spin orientation ($\boldsymbol{\sigma} \propto \mathbf{j_e} \times \mathbf{z}$) of the generated spin current is orthogonal to the out-of-plane direction (Fig. 1a), thus cannot switch the perpendicular magnetization deterministically. To solve this problem, assistance from an in-plane magnetic field has frequently been used to break the in-plane symmetry[4,5]. However, it is desirable to achieve highly efficient switching in the absence of a magnetic field.

To overcome this geometrical restriction, symmetry-reduced materials, such as transition-metal dichalcogenides[11] and non-collinear antiferromagnets[12,13], have been proposed as candidate materials whose unique crystal structure and spin texture break the in-plane symmetry. Some of them have demonstrated a field-free switching behavior[14]. However, these materials require single-crystal structures epitaxially grown on special substrates and are therefore incompatible with device applications. Apart from such symmetry-reduced materials, FMs (such as Co, Fe, and Ni) have a great potential to lift the geometrical restriction through time-reversal asymmetry due to the presence of magnetization[15–21]. From a symmetry point of view, the spin orientation of the out-of-plane flowing spin current in FM can be expressed as follows (Fig. 1b, c):

$$\boldsymbol{\sigma}_{\mathbf{MI}} \propto \mathbf{j_e} \times \mathbf{z},$$

$$\boldsymbol{\sigma}_{\mathbf{MD}} \propto \mathbf{m} \times (\mathbf{j_e} \times \mathbf{z}), \qquad (1)$$

where, $\mathbf{m}$ is the unit magnetization vector of FM. There are two types of spin current with different spin-orientation symmetries, $\boldsymbol{\sigma}_{\mathbf{MI}}$ and $\boldsymbol{\sigma}_{\mathbf{MD}}$, where MI and MD are the magnetic-independent and magnetic-dependent components, respectively. As in the case of NMs, the former has the restriction mentioned above. On the other hand, the latter is free from the restriction, because its spin orientation can be controlled by the magnetization direction. Therefore, SOT with $\boldsymbol{\sigma}_{\mathbf{MD}}$ is of great interest, because it enables switching of perpendicular magnetization. However, there have been very few reports on the generation of $\boldsymbol{\sigma}_{\mathbf{MD}}$[15–18] and the efficiency of spin-current generation, characterized by spin-current conductivity, has been low in FMs. To overcome this issue and achieve high spin-current conductivity, it is necessary to clarify the physical origin of the charge-to-spin conversion in FM, for instance, whether the interfacial contribution or bulk contribution is dominant.

In this study, we report on the enhancement of spin-current conductivity for magnetic-dependent charge-to-spin conversion in FM by clarifying its origin. We show the first experimental evidence for the coexistence of interfacial and bulk contributions to the generation of $\boldsymbol{\sigma}_{\mathbf{MD}}$ through layer thickness dependence. We also report up to a threefold increase in the interfacial contribution by using interfacial band engineering. We achieved a giant spin-current conductivity of $\boldsymbol{\sigma}_{\mathbf{MD}}$ on the order of $10^2\,\Omega^{-1}\,\mathrm{cm}^{-1}$, an order higher than that of other symmetry-reduced materials.

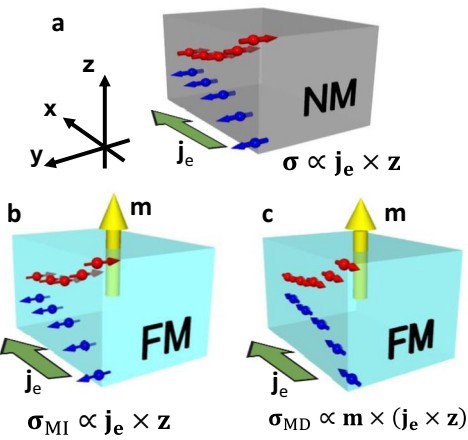

**Fig. 1 Charge-to-spin conversion in NMs and FMs. a** Charge current-induced spin current in NMs. $\boldsymbol{\sigma}$ is defined as spin polarization of the generated spin current. Green arrow shows the direction of charge current $\mathbf{j_e}$. **b, c** Charge current-induced spin current in FMs. Yellow arrow shows magnetization of FM ($\mathbf{m}$). Two spin currents with different spin polarization can be generated: **b** $\boldsymbol{\sigma}_{\mathbf{MI}}$ and **c** $\boldsymbol{\sigma}_{\mathbf{MD}}$.

## Results

**Spin-current generation from ferromagnet**. As a platform for investigating charge-to-spin conversion in ferromagnets, we used a PML/Cu/IML trilayer structure, where PML and IML stand for perpendicularly magnetized and in-plane magnetized ferromagnetic layers, respectively, as shown in Fig. 2a. The PML acts as a spin-current source layer and the IML acts as a free layer to detect the spin injection from PML. It is noteworthy that the spin anomalous Hall effect[22] can be excluded from the present structure, because the spin current generated by this effect flows in the plane when the FM is perpendicularly magnetized. Therefore, PMLs are ideal spin-current sources with which to quantitatively investigate $\boldsymbol{\sigma}_{\mathbf{MD}}$. Cu was chosen as a non-magnetic spacer layer, because it exhibits a negligibly small spin Hall effect and relatively long spin diffusion length[23]. We used a 1.3 nm-thick $Fe_{75}B_{25}$ (Fe-B) layer as the IML and a Co/Ni multilayer, which is a typical FM with strong perpendicular magnetic anisotropy[24], as the PML. We prepared three series (Series A, B, and C) of differently prepared PMLs, the details of which are explained in "Methods."

The charge-to-spin conversion was characterized by conducting spin-torque ferromagnetic resonance (ST-FMR) measurements under an in-plane DC bias current ("Methods"). Resonance spectra were obtained by sweeping the external magnetic field $H_{\mathrm{ext}}$ in the $xy$ plane, as shown in Fig. 2b. Here, the resonance spectra originate from the ferromagnetic resonance in the IML, which was confirmed from the Kittel plot and magnetoresistance curves (Supplementary Figs. 1 and 5). Figure 2c shows examples of the modulated magnetic Gilbert damping constant $\Delta\alpha$ with the PML magnetized along the $+\mathbf{z}$ and $-\mathbf{z}$ direction (red and blue data, respectively) as a function of bias current $I_{\mathrm{DC}}$. Hereafter, we refer to these states as $+M_{\mathrm{PML}}$ and $-M_{\mathrm{PML}}$, respectively. The clear linear modulation of $\Delta\alpha$ can be explained by considering the spin-current injection from the PML, as described below. The DC charge current in the PML generates a spin current flowing in the out-of-plane direction due to the charge-to-spin conversion. When this spin current is injected into the IML, spin torque acts on the magnetic moment of the IML, resulting a linear response[6]. In addition, the modulation efficiency $\Delta\alpha/I_{\mathrm{DC}}$, which corresponds to the slope of $I_{\mathrm{DC}}$ vs. $\Delta\alpha$ in Fig. 2c, differs between two magnetization states

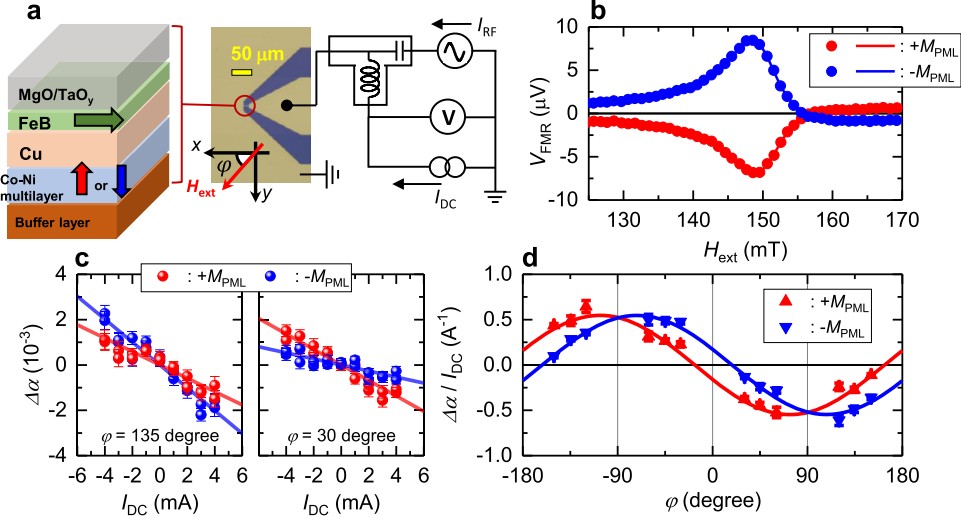

**Fig. 2 Trilayer-based structure and spin-torque ferromagnetic resonance. a** Schematic of multilayer stacks (left) and setup of ST-FMR measurement (right). **b** Resonance spectra of $+M_{PML}$ and $-M_{PML}$ states (red and blue data, respectively). $\mu_0$ is vacuum permeability. Solid curves show fitting curve using Eq. (3). **c** Dependence of modulated Gilbert damping coefficient $\Delta\alpha$ on applied bias current $I_{DC}$ under $+M_{PML}$ (red) and $-M_{PML}$ states (blue). Left and right graphs show measurement under $\varphi = 135°$ and $30°$, respectively. Solid lines show linear fitting results. **d** Angular dependence of damping modulation efficiency $\Delta\alpha/I_{DC}$ and $\pm M_{PML}$ state (red and blue data, respectively). Solid curves show fitting results using Eq. (2).

of the PML and the relationship of slope difference reverses at in-plane angle $\varphi = 90°$. These features indicate the presence of magnetic-dependent spin current in our trilayer structure.

Figure 2d shows the full angular dependence of $\Delta\alpha/I_{DC}$ in the $xy$ plane. The $\Delta\alpha/I_{DC}$ is proportional to the amount of spin injected into the IML and projected to the magnetization in the IML. In the present geometry, where the PML is magnetized along the z-axis, $\sigma_{MI}$ and $\sigma_{MD}$ are oriented along the y- and the x-axes, respectively. Therefore, we can quantitatively estimate both the magnitude and orientation of injected spin. Taking $\sigma_{MI}$ and $\sigma_{MD}$ in the present system into account, the angular dependence of $\Delta\alpha/I_{DC}$ can be expressed as

$$\Delta\alpha/I_{DC} = c[-\xi_{MI} \sin\varphi + \text{sgn}(m_z^{PML})\xi_{MD}\cos\varphi], \quad (2)$$

where, $\xi_{MI(MD)}$ represents the spin-current conductivity of the magnetic-independent(-dependent) charge-to-spin conversion, $m_z^{PML}$ is the **z**-component of magnetization of PML, and $c$ is the coefficient ("Methods"). For both $+M_{PML}$ and $-M_{PML}$ states, the plots are well fitted to Eq. (2) (solid curves) in Fig. 2d. From the fitting curves, we obtain $\xi_{MI}$ and $\xi_{MD}$ of $(0.60 \pm 0.02) \times 10^3 \ \Omega^{-1}\text{cm}^{-1}$ and $(-0.25 \pm 0.01) \times 10^3 \ \Omega^{-1}\text{cm}^{-1}$, respectively. The sign of $\xi_{MI}$ in present system is the same as that of the spin Hall effect in Pt[6,7], whereas the sign of $\xi_{MD}$ is consistent with that of the CoFe/Ni and Co/Ni multilayer systems[15,25] reported previously. We note that the IML (Fe-B layer) may also generate spin currents, resulting in a torque acting on PML. However, this torque is negligible in our system. The reason for this is that the spin current generated from IML is relatively small compared to PML due to high resistivity of Fe-B layer. In addition to this, high perpendicular magnetic anisotropy of PML makes it difficult to induce magnetization dynamics in PML.

In addition to the ST-FMR measurements, we conducted two different experiments to confirm the existence of $\xi_{MI}$ and $\xi_{MD}$ in the present system. One experiment is the spin pumping measurement[26], in which we focused on the electromotive force originated from reciprocal process of the charge-to-spin conversion (Supplementary Note 8). We confirmed that the charge-to-spin conversion induced electromotive force is well explained by both $\xi_{MI}$ and $\xi_{MD}$ obtained from our ST-FMR measurement (Supplementary Fig. 10). The second experiment is the field-free

magnetization switching measurement. This was motivated by the feature of magnetic-dependent charge-to-spin conversion, which can generate an out-of-plane polarized spin current by arranging the magnetization of the source layer parallel (or anti-parallel) to the charge current direction (Supplementary Note 9). Moreover, the vector of the out-of-plane spin polarization reverses with reversing the magnetization of spin source layer. To confirm these features, we prepared an additional trilayer structure that consists of an in-plane magnetized source layer and perpendicularly magnetized free layer. We obtained chiral switching behavior in the absence of an external field (Supplementary Fig. 11). These results support the ST-FMR experiment revealing the existence of $\xi_{MI}$ and $\xi_{MD}$ in ferromagnets.

**Interfacial and bulk contributions to charge-to-spin conversion.** To investigate the underlying physics of the observed charge-to-spin conversion, we studied the dependence of $\xi_{MI}$ and $\xi_{MD}$ on PML thickness by using the Series A and B samples. From the dependence on the spin source layer thickness, we can determine the contributions from the bulk and interfacial origins. For the bulk contribution, a significant thickness dependence should be observed. In the case of the bulk mechanism, such as the spin Hall effect in NMs, e.g., its magnitude increases with NM thickness and saturates, which is determined by spin diffusion length. In the interfacial mechanism, on the other hand, the thickness dependence should saturate much more rapidly compared with the bulk mechanism, because the spin current is generated only at the PML/Cu interface. Figure 3a shows the dependence of the two conversion efficiencies, $\xi_{MI}$ and $\xi_{MD}$, on PML thickness $t_{PML}$. Both efficiencies show significant thickness dependence but with different trends, where $|\xi_{MI}|$ increases with $t_{PML}$ and $|\xi_{MD}|$ is suppressed by $t_{PML}$.

We now discuss the origin of the thickness dependence of $\xi_{MI}$ and $\xi_{MD}$. The monotonic increase in $\xi_{MI}$ indicates a bulk effect originating from the spin Hall effect in the PML. The spin-current conductivity obtained here is a positive value, which is consistent with the spin Hall effect in Ni[27,28]. Therefore, this spin Hall effect in the PML should mainly originate from the spin Hall effect of Ni. It is noteworthy that the spin Hall effect from the buffer layer, such as the 1.5 nm-thick Ir layer[29], cannot explain

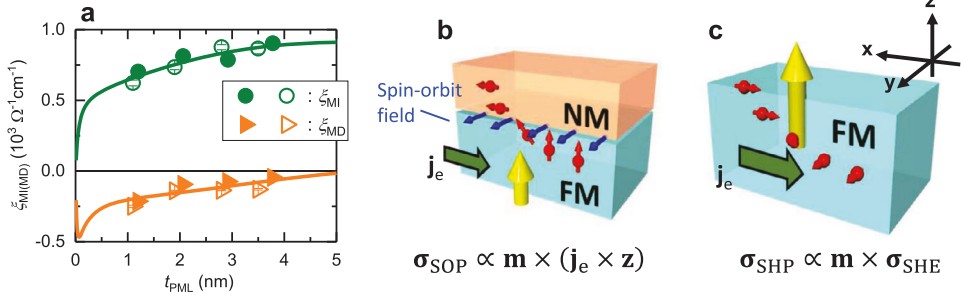

**Fig. 3 Origin of charge-to-spin conversion in FM. a** Dependence of $\xi_{MI}$ and $\xi_{MD}$ on PML thickness in Series A (filled symbol) and B (open symbol) samples. Solid curve shows the fitting results using theoretical formula. **b, c** Schematic of two generation mechanisms of $\boldsymbol{\sigma}_{MD}$. Yellow arrow indicates magnetization of FM (**m**) and red arrow is spin orientation of spin current. **b** Spin–orbit precession effect triggered by spin–orbit field (blue arrow) at the interface ($\boldsymbol{\sigma}_{SOP}$). **c** Proposed spin-Hall precession effect within FM ($\boldsymbol{\sigma}_{SHP}$).

the monotonic increase of $\xi_{MI}$. In addition, it is worth noting that the interfacial contribution of $\boldsymbol{\sigma}_{MI}$, such as spin–orbit filtering effect[16,20] can generate sizable $\xi_{MI}$. In this study, neither sign reversal nor a significant drop in $\xi_{MI}$ was observed. Therefore, $\xi_{MI}$ in our system may originate from the spin–orbit filtering effect and spin Hall effect with the same polarity. On the other hand, in the case of $\xi_{MD}$, which is an $\mathbf{m} \times (\mathbf{j}_e \times \mathbf{z})$ oriented spin, the magnitude of the spin-current conductivity increases as the PML thickness is reduced. This significant thickness dependence cannot be explained by the interfacial contribution[16]. Therefore, the thickness dependence provides evidence of the existence of an additional bulk contribution with opposite signs to the interfacial contribution. This was also observed in another system using Co/Pt multilayers as a PML (Supplementary Note 7).

We discuss the origin of each contribution to the magnetic-dependent charge-to-spin conversion. First, the interfacial contribution can be explained by the presence of the spin–orbit precession effect at the PML/Cu interface, as proposed by Baek et al.[16] and Amin et al.[20] Figure 3b illustrates the concept of the spin–orbit precession effect. When the incoming spin carrier in FM is scattered at the interface between the FM and NM, spins orthogonal to the spin–orbit field (blue arrow) rotate about the spin–orbit field. When this spin carrier is polarized as in the case of FM, a finite spin current oriented along the $\mathbf{m} \times (\mathbf{j}_e \times \mathbf{z})$ direction is generated. It is noteworthy that $\mathbf{j}_e \times \mathbf{z}$ reflects the vector of the spin–orbit field derived from Rashba–Edelstein effect. Next, we discuss the bulk contribution. Multiple mechanisms have been proposed as an origin of the bulk contribution on $\boldsymbol{\sigma}_{MD}$. For instance, Pauyac et al.[30] proposed a spin swapping effect, which generates an $\mathbf{m} \times \mathbf{y}$ polarized spin current within a diffusive ferromagnet[30]. However, this effect is considered to be weaker in FM with strong exchange interaction, such as Co/Ni and Co/Pt multilayers. Another candidate that generates the $\mathbf{m} \times \mathbf{y}$ polarized spin current is the extrinsic mechanism, which is related in origin to the planar Hall effect[20]. In addition to these effects, we propose an alternative mechanism to reproduce our layer thickness dependence of $\xi_{MD}$. Here we focused on the mechanism originating from the combination of the spin Hall effect within FM and its interaction with local magnetization (Fig. 3c). As mentioned above, $\boldsymbol{\sigma}_{MI}$ is generated via the spin Hall effect in the PML. As $\boldsymbol{\sigma}_{MI}$ is orthogonal to the magnetization of the PML, it precesses around the magnetization through the exchange interaction[31,32]. As a result, not only the ordinary spin current induced by the spin Hall effect (we define as spin Hall current with polarization $\boldsymbol{\sigma}_{SHE} \propto \mathbf{j}_e \times \mathbf{z}$ but also an alternative spin current with an $\mathbf{m} \times \boldsymbol{\sigma}_{SHE}$ spin orientation is generated. This spin current strongly depends on spin dephasing length $\lambda_J$. When $\lambda_J$ is too short (a few Å), $\boldsymbol{\sigma}_{SHE}$ immediately vanishes before $\mathbf{m} \times \boldsymbol{\sigma}_{SHE}$ spin orientation is generated. However, when $\lambda_J$ is relatively long

$(1 \sim 2 \text{ nm})$[31], which has been experimentally demonstrated by several groups[33,34], $\mathbf{m} \times \boldsymbol{\sigma}_{SHE}$ spin orientation can be generated before the dephasing. This alternative charge-to-spin conversion mechanism, which we call the spin-Hall precession effect, is different from the interfacial spin–orbit precession effect, because the source is the spin Hall current in the spin-Hall precession effect, whereas it is the spin-polarized charge current in the spin–orbit precession effect. In addition, the spin precesses about the magnetization via the exchange interaction for the spin Hall precession, whereas an interfacial spin–orbit field takes place for the spin–orbit precession effect. In this mechanism, the precession direction of the spin Hall current also reverses with reversing the magnetization, resulting in sign reversal of spin polarization with no difference in magnitude between $+M_{PML}$ and $-M_{PML}$ states (Supplementary Fig. 8). In the present system, the spin-orientation vector of the proposed spin-Hall precession effect is positive ($\xi_{MD} > 0$), as the spin Hall angle of the PML is positive, from $\xi_{MI}$ in Fig. 3a. We derived a theoretical formula describing the thickness dependence of $\xi_{MI}$ and $\xi_{MD}$ using a theory[32,33] combining the diffusive spin transport theory[31] in bulk and the spin-dependent Landauer theory of interface transport[35] (Supplementary Note 5). We used this theoretical formula as a fitting curve in Fig. 3a and found good agreement with the experimental data on $\xi_{MI}$ and $\xi_{MD}$. The further discussion on the mechanism of bulk contribution will be shown below.

**Interface band engineering of charge-to-spin conversion.** Here we show that $\xi_{MI}$ and $\xi_{MD}$ have interfacial and bulk contributions to charge-to-spin conversion. In particular, the interfacial contribution becomes more important when the PML is in the thin region around 1 nm. We further investigated the interfacial contribution by engineering the interfacial band structure. This structure was modulated by controlling the ratio of the Co-Ni concentration within a region of 0.4 nm from the Cu spacer (Fig. 4a). The angular dependence of $\Delta\alpha/I_{DC}$ with $Co_{1-x}Ni_x$ ($x = 69\%$) near the surface is shown in Fig. 4b. Compared with the sample with only Co near the surface sample shown in Fig. 2d, the magnitude of $\Delta\alpha/I_{DC}$ near $\varphi = 0°$ and 90° was significantly enhanced, indicating an enhancement in $\xi_{MI}$ and $\xi_{MD}$ in the sample with $Co_{31}Ni_{69}$ near the surface. The dependence of $\xi_{MI}$, $\xi_{MD}$, and the absolute value ratio of $\xi_{MD}$ to $\xi_{MI}$ ($|\xi_{MD}/\xi_{MI}|$) as a function of concentration of Ni in the layer nearest to the surface is summarized in Fig. 4c. Around Ni of ~70%, $\xi_{MD}$ shows a maximum that is three times larger than that at the Co surface ($x = 0\%$). For $\xi_{MI}$, spin current conductivity is enhanced by a factor of 2 and saturates at Ni of ~25%. The $|\xi_{MD}/\xi_{MI}|$ increases up to 42% by increasing Ni concentration. This was also

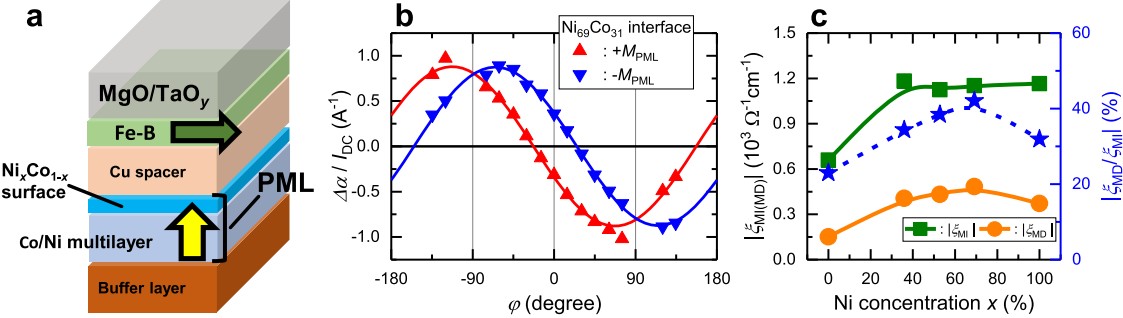

**Fig. 4 Interface band engineering of charge-to-spin conversion. a** Schematic of stack structure used for interface engineering (Series C samples). Ni concentration of the 0.4 nm-thick Co-Ni alloy layer near to the Cu interface (light blue region) is modulated. **b** Angular dependence of damping modulation efficiency $\Delta\alpha/I_{DC}$ in $Co_{31}Ni_{69}$ interface sample. Red (blue) data shows the measurement in the $+M_{PML}$ ($-M_{PML}$) state and solid curve shows fitting results using Eq. (2). **c** Ni-concentration (x) dependence of absolute value of $\xi_{MI}$ (left axis: green) $\xi_{MD}$ (left axis: orange), and $|\xi_{MD}/\xi_{MI}|$ (right axis: star symbol). Solid line is a guide for the eyes.

confirmed from the spin pumping measurement (Supplementary Note 8 and Supplementary Fig. 10).

We now discuss the origin of the strong interfacial-structure dependence. We focus on the interface-generated spin current arising from the spin–orbit filtering and spin–orbit precession effects[16,20]. Each effect generates spin current polarized along $\mathbf{j}_e \times \mathbf{z}$ and $\mathbf{m} \times (\mathbf{j}_e \times \mathbf{z})$, and contributes to $\xi_{MI}$ and $\xi_{MD}$, respectively. The following two factors play an important role in the interfacial contribution to the charge-to-spin conversion: the strength of the spin–orbit coupling induced Rashba–Edelstein effect at the interface and the spin polarization of the spin carrier. For the former, it is known that the strength of the Rashba effect is related to the work–function difference at the interface. According to Michaelson[36], the work–function difference between Ni and Cu, $\Lambda_{Ni-Cu} = 0.50$ eV, is larger than that between Co and Cu, $\Lambda_{Co-Cu} = 0.35$ eV, indicating a larger Rashba–Edelstein effect at the Ni/Cu interface. This enhancement in the Rashba–Edelstein effect in turn increases the spin–orbit filtering effect and spin–orbit precession effect at the PML/Cu interface, which explains the increase in $\xi_{MI}$ and $\xi_{MD}$ shown in Fig. 4c. The latter factor, which is the carrier-spin polarization of the FM, affects the efficiency of the spin–orbit precession effect. Previous studies have shown that the carrier-spin polarization of Co is larger than that of Ni[37]. Therefore, the carrier polarization of the $Co_{1-x}Ni_x$ layer should decrease as the Ni concentration increases. This may explain the slight drop in $\xi_{MD}$ at Ni 100%. Let us now compare our results with other materials with different mechanisms. The presence of out-of-plane polarized spin current has been reported in reduced-symmetry materials, such as $WTe_2$[11]. This phenomenon has also been observed in antiferromagnetic $NiMn_3N$, in which its non-collinear spin texture reduces the symmetry[13]. In both cases, the spin-current conductivity of the out-of-plane polarized spin is estimated to be on the order of $10 \, \Omega^{-1} \, cm^{-1}$, which is an order of magnitude smaller than $\xi_{MD}$ in our Co/Ni multilayers (on the order of $10^2 \, \Omega^{-1} \, cm^{-1}$).

## Discussion

We investigated the mechanism of charge-to-spin conversion originating from perpendicularly magnetized FM and its interface, and ascertained two important features of magnetic-dependent charge-to-spin conversion. One is that the interfacial spin–orbit precession effect plays a dominant role in achieving high spin-current conductivity. By engineering the interfacial band structure, we achieved up to threefold enhancement in magnetic-dependent charge-to-spin conversion. The other is that we experimentally show the first evidence of bulk contribution to

the magnetic-dependent charge-to-spin conversion. Whereas the interfacial contribution is considered to originate from the spin–orbit precession effect, the origin of the bulk contribution is open to discussion, because the magnetic-dependent spin current could arise from multiple mechanisms[21]. According to the first-principle calculation, extrinsic mechanism which its origin related to planar Hall effect can generate a similar spin current with $\sigma_{MD}$ symmetry[20]. Another possible source is the spin-Hall precession effect as we proposed. However, it is difficult to quantify which mechanism plays a dominant role in bulk contribution from the present experiment. Further experimental and theoretical studies are necessary to classify the detail mechanisms of the spin-current generation.

In the Co/Ni multilayer stack we used, the interfacial spin–orbit precession effect and proposed spin-Hall precession effect oppose each other, resulting in a reduction in conversion efficiency. If we can either reverse the sign of the spin–orbit precession effect or the spin Hall effect in FM, we can further enhance the efficiency of magnetic-dependent charge-to-spin conversion. For example, the sign reversal of spin–orbit precession effect can be induced by either reversing the interfacial spin–orbit field by spacer material designing[18] or using FM with negative spin polarization such as Co-Cr alloys[38]. The mechanism of spin-current generation clarified in this study would be a milestone to energy-efficient field-free switching.

## Methods

**Sample preparation.** Trilayer-based structures were deposited on a thermally oxidized silicon substrate by using an ultrahigh-vacuum magnetron sputtering system. The stacking structure was as follows: buffer layer/PML/Cu (3.0 nm)/Fe-B (1.3 nm)/MgO/TaO$_y$ (thickness in nm). PML stands for a perpendicularly magnetized Co/Ni multilayer. An Ir buffer layer was used to enhance the perpendicular magnetic anisotropy of the PML[29,39]. We confirmed that the PML showed strong perpendicular magnetic anisotropy (Supplementary Note 2 S2). The MgO/TaO$_y$ acts as a protection layer from oxidation and TaO$_y$ is formed by naturally oxidizing the 2 nm Ta layer under ambient condition. The buffer layer consists of Ta/Ru/Ir or Ta-B/Ru/Ir. We prepared three series of PMLs as shown in Table 1. The values inside parentheses are the thickness of each layer in nanometers. In Series A and B, we varied the thickness of the PML by changing the repetition number $n$ of the multilayer. The thicknesses of the Co and Ni layer was slightly changed between two series to finely change the PML thickness. In Series C, the Ni concentration $x$ of the top $Co_{1-x}Ni_x$ layer was varied from 0 to 1 to modify the interface band structure. We note that relatively sharp interface of PML/Cu and significant modulation of interface structure in series C samples were confirmed from the cross-section transmission electron microscopic image and Energy Dispersive X-ray (EDX) spectroscopy (Supplementary Note 1 S1). The films were patterned into 3 μm-wide and 12 μm-long microstrips using optical lithography and Ar ion milling. The Cr (5)/Au (200) electrodes were deposited as electrical contacts by using a lift-off process. Series A, B, and C samples were annealed at 220 °C, 240 °C, and 250 °C, respectively, for an hour after device fabrication. This annealing procedure enhanced the perpendicular magnetic anisotropy of the Fe-B/MgO interface[40], reducing the effective demagnetization field of the IML. The effective

**Table 1 Summary of the trilayer-based stack structure.**

| Series | Buffer (nm) | PML (nm) | *n* or *x* | IML (nm) |
|--------|-------------|----------|------------|----------|
| A | Ta-B(3.0)/Ru(2.0)/Ir(1.5) | Co(0.04)/[Co(0.26)/Ni(0.6)]$_n$/Co(0.3) | $n = 1$-4 | Fe-B (1.3) |
| B | Ta(3.0)/Ru(2.0)/Ir(1.5) | Co(0.3)/[Ni(0.5)/Co(0.3)]$_n$ | $n = 1$-4 | Fe-B (1.3) |
| C | Ta(3.0)/Ru(2.0)/Ir(1.5) | Co(0.4)/Ni(0.6)/Co$_{1-x}$Ni$_x$ (0.4) | $x = 0, 0.35, 0.52, 0.7, 1$ | Fe-B (1.3) |

demagnetization field of Series A, B, and C were about 116, 84, and 51 mT, respectively.

**ST-FMR measurement**. The experimental setup for the ST-FMR measurement is shown in Fig. 2a. We used a microwave signal generator (Keysight N5173B) to generate the RF current $I_{RF}$. The frequency $f$ of $I_{RF}$ was set in the range of 3.5–7 GHz and were injected into the microstrip device through the RF port of a bias-T. ST-FMR spectra were detected by measuring the DC voltage ($V_{FMR}$) arising from rectification of the magnetoresistance oscillation and the RF current[6]. A lock-in amplifier (Stanford Inst. SR830) was used to detect $V_{FMR}$ by modulating the $I_{RF}$ at a low frequency of 777 Hz. The magnetization of the PML was controlled by applying a magnetic field along the **z**-axis (the strength was set to 300 mT) before the measurement. In addition to the RF current, a DC bias current $I_{DC}$ from $-4$ to 4 mA was injected from the DC port of the bias-T to inject spin current to the IML detection layer (Fig.1a). We injected an RF current with a power of 1 dBm into the microstrip throughout the measurement. The measured $V_{FMR}$ is expressed as the sum of Lorentzian $S(H_{ext})$ and anti-Lorentzian functions $A(H_{ext})$;

$$V_{FMR} = V_A A(H_{ext}) + V_S S(H_{ext}),$$

$$S(H_{ext}) = W^2/[W^2 + (H_{ext} - H_{res})^2],$$

$$A(H_{ext}) = (H_{ext} - H_{res})S(H_{ext})/W, \qquad (3)$$

where, $H_{res}$ and $W$ corresponds to the resonance field and linewidth, respectively. $V_A$ and $V_S$ are coefficient constants that mainly originate from the giant magnetoresistance effect[25]. Details of the derivation of $V_A$ and $V_S$ are shown in Supplementary Note 4 S4. Using Eq. (3) as a fitting function, we determined the $H_{res}$ and $W$ of each spectra. The $I_{DC}$ dependence of $W$ is expressed as

$$W(I_{DC}) = W_0 + \frac{2\pi f}{\gamma}\alpha,$$

$$\alpha = \alpha_0 + (\Delta\alpha/I_{DC})I_{DC},$$

$$\Delta\alpha/I_{DC} = c[-\xi_{MI}\sin\varphi + \text{sgn}(m_z^{PML})\xi_{MD}\cos\varphi],$$

$$c = \frac{\hbar}{2e}\frac{\eta_{PML}\rho_{PML}}{wt_{PML}}\frac{1}{\mu_0 M_s t_{IML}(H_{res} + 0.5M_{demag})}, \qquad (4)$$

where $\gamma$ is the gyromagnetic ratio, $\hbar$ is the Dirac constant, $e$ is the elementary charge, $\eta_{PML}$ ($\rho_{PML}$) is the shunting current ratio (longitudinal resistivity) of the PML in the trilayer structure, $w$ is the width of the microstrip device, $M_s$ is the magnetization of the IML, $t_{PML(IML)}$ is the thickness of the PML(IML), $W_0$ is the inhomogeneous broadening of the IML, $\alpha_0$ is the Gilbert damping coefficient of the IML without bias current, and $M_{demag}$ is the effective demagnetization field of the IML. The $\alpha$, $W_0$, and $M_{demag}$ were determined from the frequency dependences of $W$ and $H_{res}$ without applying a bias current (Supplementary Fig. 1 S1).

## Data availability
The data that support the findings of this work and other findings of this study are available from the corresponding author upon reasonable request.

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

## Acknowledgements

We thank T. Nakano, E. Usuda, and M. Toyoda for technical help on the film deposition and device fabrication. We thank T. Yamamoto for technical help on the measurement setups. We thank T. Yorozu for technical help on the derivation of theoretical formula. We also thank K. J. Lee, V. P. Amin, and M. D. Stiles for valuable discussions. This work was partly supported by JSPS KAKENHI (Grant number 19J01643), JST CREST (Grant number JPMJCR18T3), Japan. Part of this work was conducted at the AIST Nano-Processing Facility, supported by the Nanotechnology Platform Program of the Ministry of Education, Culture, Sports, Science and Technology (MEXT), Japan.

## Author contributions

Y.H. conceived and designed the experiments. Y.H. deposited the trilayer-based stacks and fabricated the microstrip devices with the help of K.Y., A.F., and H.K. Y.H. carried out the measurements and analyzed the data with theoretical support from T.T. The theoretical formula was derived by T.T. Y.H. wrote the manuscript with input from T.T. and S.Y. All authors discussed the results.

## Competing interests

The authors declare no competing interests.
