## [Transparent Peer Review file · Nature Communications]

Reviewer comments & decisions:

Reviewer comments, first version:

REVIEWER COMMENTS

Reviewer #1 (Remarks to the Author: Overall significance):

The authors have used ST-FMR method to find out the origin of spin torque rising from FM/NM structure. This is a quite interesting research and provide some insight that may enhance SOT efficiency and possibly field-free switching with FM layers. I think the results are quite reasonable and well organized.

Reviewer #1 (Remarks to the Author: Strength of the claims):

1. I suppose the main reason for Fig.2(c) is to show that the slope of $-MPML$ and $+MPML$ are different, which is one of reasons that supports the existence of magnetization dependent torque. Then, I think the authors need to find out a good way to show that the slopes of blue and red lines are different. In current configuration, it is difficult to know if there are any differences between the two lines.
2. In Fig.3(a), why are Series A and B fitted with a same line? Shouldn't there be separate fitting lines for Series A and B?
3. With Co/Ni or Co/Pt, the authors showed that the sign of spin-orbit precession and the spin hall precession effect compete with each other. Any suggestions on how to reverse the sign of the spin-orbit precession effect or the spin Hall effect in FM in order to enhance the efficiency?

Reviewer #1 (Remarks to the Author: Reproducibility):

1. I think a concrete evidence of spin hall precession effect would be to show a difference in spin torque effect between $-MPML$ and $+MPML$. Can the authors provide this evidence? Or do the authors think Figure 2(a) supports this phenomenon?
2. The authors did 'some' experiments to verify the origin of the torques rising from FM layer. Basically

current flowing in x-direction in FM materials with z-magnetization can provide both y-direction spin orientation (MI) and x-direction spin orientation (MD). I think this is an interesting paper, that gives some new approach to field free switching and highly efficient SOT devices.

Unfortunately, the experiment only considers ST-FMR as the evaluation method. If there were another method like current induced SOT switching to cross-check their experimental data, then the paper may be suitable for journals like Nature Communications (It does not necessarily have to be SOT switching, just another different method to cross-check). But in current situation, I think Communications Physics is the suitable journal for publication, if the authors can give good answers to my comments.

Reviewer #2 (Remarks to the Author: Overall significance):

The authors perform ST-FMR measurements in ferromagnetic trilayers to probe unconventional spin current generation, both from the fixed ferromagnetic layer and from the interface between that layer and the nonmagnetic spacer layer. By varying the thickness of the fixed layer (with perpendicular magnetization) and the Co-Ni concentration near the interface, the strength of the bulk and interfacial contributions to the torques are studied. The unconventional torques are an order of magnitude larger than those reported in other systems that break the required mirror symmetries, such as transition-metal dichalcogenides and non-collinear antiferromagnets. The authors then propose a phenomenological mechanism by which spin Hall currents rotate their spin direction about the magnetization in ferromagnets to explain the bulk-originating contribution to the unconventional ($m \times j \times z$) torques. The manuscript is fairly well written, and the results are clearly described in the main text and figures. One of the main contributions beyond previous work is the study of interfacial composition on the unconventional torques and (to a lesser extent) the proposed theoretical mechanism. Given a few revisions, I believe this work warrants publication in one of the suggested Nature journals. I have a few comments that I list below:

Comments

1) The mechanism suggested for bulk spin current generation in ferromagnets, called the “spin Hall precession effect” by the authors, is supported by drift-diffusion calculations but not the only plausible mechanism. It is simpler and more accurate to say that the bulk-originating spin current is allowed by symmetry in ferromagnets and could arise from multiple mechanisms, since it is unclear which mechanism dominates. Alternate mechanisms have already been suggested by other researchers (aside from spin swapping, which is mentioned in the manuscript). For instance, in Phys. Rev. Lett. 121, 136805 (which the authors already cite regarding interface-generated spin currents), a bulk mechanism is introduced that generates “spin-orbit precession-like” currents in infinite Co leads, supported by first principles transport calculations. The authors state that the origin of this effect is related to the planar Hall effect. Furthermore, various authors have reported on the so-called “magnetic spin Hall effect”

(Phys. Rev. Research 2, 023065) which allows for spin currents of the same orientation as discussed above. Note that the spin currents discussed in both references have a similar underlying origin. Thus, given that there are several pathways for bulk spin currents to exist with spin direction along $\mathbf{m} \times (\mathbf{j} \times \mathbf{z})$ (with some higher-order contributions), it would seem best to refrain from singling out a particular bulk mechanism. The authors could certainly suggest that the “spin Hall precession effect” is a candidate but should inform the readers of other possibilities.

2) The motivation of this paper is the control of out-of-plane magnetizations, yet in this work the source layer has an out-of-plane magnetization and the free layer has an in-plane magnetization. The authors should at least comment on why this orientation was chosen. There are two cases of interest: 1) z-magnetization, x-electric field, z-spin flow, x-spin direction and 2) x-magnetization, x-electric field, z-spin flow, z-spin direction. Both cases are allowed by symmetry in ferromagnets. The first case is studied in this experiment whereas the latter is more useful for controlling magnetization dynamics in PML films. Certainly, both cases are related by the parameter σ_{MD} , but given higher-order-in-magnetization contributions, the spin currents in each of these cases could be different. It would be useful for the authors to at least discuss how the measurements done in this geometry will help optimize the control of out-of-plane magnetizations in other geometries.

Reviewer #2 (Remarks to the Author: Impact):

I believe this paper provides useful insight into the various spin current generation mechanisms in ferromagnetic trilayers.

Reviewer #2 (Remarks to the Author: Strength of the claims):

Overall, I consider the results and conclusions to be fairly convincing. However, it is worth raising an additional concern. Thickness-dependent measurements are one of the few ways to isolate interfacial contributions. However, since they require measurements to be performed over different samples, it is difficult to ensure that each sample has a similar interface, and that only the thickness of one layer is significantly different. However, given the lack of better approaches, it is worth performing these experiments. The results could be strengthened if more was said about the quality of the interfaces in each of the samples.

Reviewer #3 Remarks to the Author:

See attachment

In this manuscript, the authors demonstrate strong charge-to-spin conversion in ferromagnet. They utilize Co/Ni multilayers as the spin source, which shows bulk perpendicular magnetic anisotropy and benefits the thickness dependent analysis. This configuration also excludes the spin anomalous Hall effect as the origin. Both theoretical calculation and experimental results are concise, and the giant charge-to-spin conversion efficiency, which is an order of magnitude larger than the materials with reduced symmetry such as WTe₂, is exciting. However, the mechanism is still not clarified clearly enough as following, therefore I suggest this work to be further evaluated by *Communication Physics*.

1. In abstract, it is declared that “The interfacial contribution to charge-to-spin conversion is found to be dominant”. However, this is only true when the Co/Ni multilayer is thin. The bulk contribution is also non-negligible from the thickness dependence result.

2. The detailed mechanism of bulk spin-Hall precession is not explained clearly. Why such spin would not undergo dephase? Why it would precess from one orthogonal-to-magnetization direction to the other?

3. The spin-orbit precession is still dominant for ξ_{MD} in this system. I don't understand why they could realize such a large efficiency. Besides, this comparison is based on the materials with reduced symmetry, but the mechanisms are totally different. What if compare to the similar ferromagnetic trilayers?

4. In Fig. 3a, magnetic dependent spin current conductivity decreases with the increase of the PML thickness. But the theoretical fitting shows an opposite tendency when the PML thickness is very small. How do you explain this divergency between calculating and experiment? Since the interface and bulk have opposite contribution to the charge-to-spin conversion, is it possible to manipulate the sign of magnetic

dependent spin current conductivity by further increase the PML layer thickness?

5. The authors claim that both the Rashba effect and the spin polarization contribute to the interface-related charge-to-spin conversion, but they did not explain why. If these two effects are linear with the Ni concentration and have opposite sign, why ξ_{MD} drop at Ni 100%?

6. The authors should clarify that the solid lines in Fig. 4c are just guide to the eye.

Author rebuttal, first version:

Replies to Reviewers' Comments

GUIDEDO A-21-00047

Giant charge-to-spin conversion in ferromagnet via spin-orbit coupling

By Y. Hibino, T. Taniguchi, K. Yakushiji, A. Fukushima, H. Kubota, and S. Yuasa

First, we would like to thank all the reviewers for the careful reading of our manuscript and for his/her valuable comments that helped us to improve the original manuscript. Here, our responses to the referees' comments are shown below. In the attached marked up manuscript, the revised part is marked with yellow.

► **Referee #1 (Comments to author):**

The authors have used ST-FMR method to find out the origin of spin torque rising from FM/NM structure. This is a quite interesting research and provide some insight that may enhance SOT efficiency and possibly field-free switching with FM layers. I think the results are quite reasonable and well organized.

► **Our reply:**

We thank the reviewer for his/her positive evaluations.

► **Referee #1 (Technical Comments to author):**

I suppose the main reason for Fig.2(c) is to show that the slope of $-M_{\text{PML}}$ and $+M_{\text{PML}}$ are different, which is one of reasons that supports the existence of magnetization dependent torque. Then, I think the authors need to find out a good way to show that the slopes of blue and red lines are different. In current configuration, it is difficult to know if there are any differences between the two lines.

► **Our reply:**

We thank the reviewer for the suggestion to make the figure easier to understand for readers. Accordingly, we revised the Fig. 2c which are shown in Fig. R1. We revised the Fig. 2c as following 2 points:

- 1) We integrated the $+M_{\text{PML}}$ and $-M_{\text{PML}}$ under $\varphi = 135$ degree in one figure to clearly show the slope difference of two states. In addition to this, we changed the vertical axis of Figure 2c from effective damping coefficient α to modulated effective damping coefficient $\Delta\alpha$. This modification was made to emphasize the visibility of the slope difference between $+M_{\text{PML}}$ and $-M_{\text{PML}}$ states.
- 2) We added an extra data under in-plane field angle of $\varphi = 30$ degree. In this angle, the slope difference between two states reverses compared to $\varphi = 135$ degree. This would be a good way to show the clear difference of two states and supports the existence of magnetization dependent torque acting on the system.

Accordingly, we have revised the main text as following:

<Spin current generation from ferromagnet, Page 3 line 11 ~ 14>

"In addition, the modulation efficiency $\Delta\alpha/I_{\text{DC}}$, which corresponds to the slope of I_{DC} vs. $\Delta\alpha$ in Fig. 2c, differs between two magnetization states of the PML, and the relationship of slope difference

reverses at in-plane angle $\varphi = 90$ degree. These features indicate the presence of magnetic-dependent spin current in our tri-layer structure.”

Figure R1: Tri-layer-based structure and spin torque ferromagnetic resonance (a) Schematic of layer stacks (left) and setup of the ST-FMR measurement (right). (b) Resonance spectra of $+M_{PML}$ and $-M_{PML}$ states (red and blue data respectively) μ_0 is the permeability under vacuum. The solid curves show the fitting curve using equation (3). (c) Dependence of modulated Gilbert damping coefficient $\Delta\alpha$ on applied bias current I_{DC} under $+M_{PML}$ (red) and $-M_{PML}$ states (blue). The left and right graph shows the measurement under $\varphi = 135$ and 30 degree respectively. The solid lines show linear fitting results. (d) Angular dependence of damping modulation efficiency $\Delta\alpha / I_{DC}$ under $\pm M_{PML}$ state (red and blue data respectively). The solid curves show fitting results using equation (2).

► **Referee #1 (Comments to author):**

In Fig.3(a), why are Series A and B fitted with a same line? Shouldn't there be separate fitting lines for Series A and B?

► **Our reply:**

We thank the reviewer for the comment. The difference between Series A and B is the thickness of the Co and Ni layer. The propose of this is to change the PML thickness. Except for the PML thickness, both series are almost the same structure (the Ta and Ta-B buffer layer differs but it does not affect the result due to its high resistivity) and the resistivity of PML is same between two systems. Thus, we think that there is no problem with fitting both series with same fitting curve.

In the revised manuscript, we added the following sentence to mention the purpose for preparing

Series A and B samples.

<Methods: Sample preparation, Page 7 line 17~18>

“The thickness of Co and Ni layer is slightly changed between two series to finely change the PML thickness.”

► **Referee #1 (Comments to author):**

With Co/Ni or Co/Pt, the authors showed that the sign of spin-orbit precession and the spin Hall precession effect compete with each other. Any suggestions on how to reverse the sign of the spin-orbit precession effect or the spin Hall effect in FM in order to enhance the efficiency?

► **Our reply:**

We thank the referee for pointing out this point. There are two possible approaches to reverse the sign of the spin-orbit precession effect. One is reversing the interface spin-orbit field which can greatly be modulated by the non-magnetic spacer materials. For example, Amin *et.al* have demonstrated the sign reversal of spin-orbit precession effect in Co between Cu spacer and Pt spacer system [Amin *et.al*, Phys. Rev. Lett. **121**, 136805 (2018)]. This was also experimentally demonstrated recently [Y. Hibino *et.al*, APL Mater. **8**, 041110 (2020)]. Second is using ferromagnetic materials with negative spin polarization such as Co-Cr alloys [Vouille *et.al*, Phys. Rev. B **60**, 6710 (1999)]. In the case of spin Hall effect in ferromagnetic material, first principle calculations show that both magnitude and sign of spin Hall effect strongly vary with ferromagnetic material according to both intrinsic mechanism [Amin *et.al*, Phys. Rev. B **99**, 220405(R) (2020)].

In the revised manuscript, we added the following sentence on possible method to reverse the spin-orbit precession effect of the system:

<Discussion, Page 6 line 35 ~ Page 7 line 2~4>

“If we can either reverse the sign of the spin-orbit precession effect or the spin Hall effect in FM, we can further enhance the efficiency of magnetic-dependent charge-to-spin conversion. For example, sign reversal of spin-orbit precession effect can be induced by either reversing the interfacial spin-orbit field by spacer material designing [17] or using ferromagnetic material with negative spin polarization such as Co-Cr alloys [36].”

► **Referee #1 (Comments on reproducibility to author):**

I think a concrete evidence of spin Hall precession effect would be to show a difference in spin torque effect between $-M_{\text{PML}}$ and $+M_{\text{PML}}$. Can the authors provide this evidence? Or do the authors think Figure 3(a) supports this phenomenon?

► **Our reply:**

We thank the reviewer's valuable comment. In the spin Hall precession effect scheme, the spin Hall current with $-y$ spin polarized rotates about the magnetization resulting $+x$ ($-x$) direction under $+M_{\text{PML}}$ ($-M_{\text{PML}}$) state. This is because the rotation direction reverses with reversing the magnetization of PML. Therefore, there would be no spin torque efficiency difference between $+M_{\text{PML}}$ ($-M_{\text{PML}}$) states. To show this, we show the ζ_{MD} under $-M_{\text{PML}}$ and $+M_{\text{PML}}$ state in series A samples in Fig. R2 which show negligible difference in absolute value between two states.

According to the reviewer's comment, we added the following sentence to the revised manuscript. In addition, we added the Fig. R2 to the supplementary information S6.

<Interfacial and bulk contribution to the charge-to-spin conversion, Page 5, line 17~19>

"In this mechanism, the precession direction of the spin Hall current also reverses with reversing the magnetization, resulting in sign reversal of spin polarization with no difference in magnitude between $+M_{\text{PML}}$ and $-M_{\text{PML}}$ state (Supplementary Information S6)."

Figure R2. (a) thickness dependence of spin current conductivity ζ_{MI} (green) and ζ_{MD} (orange) under $+M_{\text{PML}}$ and $-M_{\text{PML}}$ state (filled and open symbol respectively). (b) thickness dependence of absolute value of ζ_{MD} under $+M_{\text{PML}}$ and $-M_{\text{PML}}$ state. The data is obtained from series A samples.

► **Referee #1 (Comments on reproducibility to author):**

The authors did “some” experiments to verify the origin of the torques rising from FM layer. Basically, current flowing in x-direction in FM materials with z-magnetization can provide both y-direction spin orientation (MI) and x-direction spin orientation (MD). I think this is an interesting paper, that gives some new approach to field free switching and highly efficient SOT devices.

Unfortunately, the experiment only considers ST-FMR as the evaluation method. If there were another method like current induced SOT switching to cross-check their experimental data, then the paper may be suitable for journals like Nature Communications (it does not necessarily have to be SOT switching, just another different method to cross-check). But in current situation, I think Communications Physics is the suitable journal for publication, if the authors can give good answers to my comments.

► **Our reply:**

We thank the reviewer for the suggestion for the cross-check of the experimental result. For the cross-check, we conducted additional experiments and confirmed the consistency with our ST-FMR results shown in the main text. In the following, we show the abstract of the additional experiments. These experiments are added in the supplementary information section S7 and S8.

- 1) we demonstrate the inverse process of charge-to-spin conversion, i.e., spin-to-charge conversion which arise from the Onsager’s reciprocity to check the robusticity of our observed results. We used spin pumping method to inject pure spin current to the Co/Ni multilayers and measured the spin-to-charge conversion induced electromotive force. As a result, we observed a clear electromotive force originates from the both magnetization-independent and magnetization-dependent spin-to-charge conversion in Co/Ni multilayer. In addition, we confirmed that interface structure dependence of torque ratio ($|\xi_{MD}/\xi_{MI}|$) measured by spin-pumping method reproduces the ST-FMR measurement result.
- 2) According to our results, we can generate out-of-plane spin polarized spin current by changing the magnetization configuration of the spin source ferromagnetic material from perpendicularly magnetized to in-plane magnetized system. To confirm this, we carried out additional experiments to check this through SOT-induced magnetization switching. We prepared additional tri-layer system consist from two ferromagnetic layer, which are in-plane magnetized spin source layer and out-of-plane magnetized free layer, and magnetization of free layer was probed through Hall measurement. As a result, we observed a SOT-induced chiral switching behavior strongly which its chirality reverses with the magnetization configuration of in-plane

magnetized spin source layer. The polarity of the chiral switching is well explained by the magnetization-dependent charge-to-spin conversion that we observed from ST-FMR measurement.

We hope our response and additional experiments to the reviewer's comment would resolve the reviewer's concern and take our revised manuscript into account for publication in Nature Communications.

► Referee #2 (Comments to author):

The authors perform ST-FMR measurements in ferromagnetic trilayers to probe unconventional spin current generation, both from the fixed ferromagnetic layer and from the interface between that layer and the nonmagnetic spacer layer. By varying the thickness of the fixed layer (with perpendicular magnetization) and the Co-Ni concentration near the interface, the strength of the bulk and interfacial contributions to the torques are studied. The unconventional torques are an order magnitude larger than those reported in other systems that break the required mirror symmetries, such as transition-metal dichalcogenides and non-collinear antiferromagnets. The authors then propose a phenomenological mechanism by which spin Hall currents rotate their spin direction about the magnetization in ferromagnets to explain the bulk-originating contribution to the unconventional $m \times (j \times z)$ torques. The manuscript is fairly well written, and the results are clearly described in the main text and figures. One of the main contributions beyond previous work is the study of interfacial composition on the unconventional torques and (to a lesser extent) the proposed theoretical mechanism. Given a few revisions, I believe this work warrants publication in one of the suggested Nature journals. I have a few comments that I list below

► Our reply:

We thank the reviewer for pointing out the novelty and importance of our work and support the publication of our manuscript. We also thank the reviewer for summarizing the critical points of our work.

► Referee #2 (Comments to author):

The mechanism suggested for bulk spin current generation in ferromagnets, called the "spin Hall precession effect" by the authors, is supported by drift-diffusion calculations but not the only plausible mechanism. It is simpler and more accurate to say that the bulk-originating spin current is allowed by symmetry in ferromagnets and could arise from multiple mechanisms, since it is unclear which mechanism dominates. Alternate mechanisms have already been suggested by other researchers (aside from spin swapping, which is mentioned in the manuscript). For instance, in Phys.

Rev. Lett. 121, 136805 (which the authors already cite regarding interface-generated spin currents), a bulk mechanism is introduced that generates “spin-orbit precession-like” currents in infinite Co leads, supported by first principle calculations. The authors state that the origin of this effect is related to the planar Hall effect. Furthermore, various authors have reported on the so-called “magnetic spin Hall effect” (Phys. Rev. Research 2, 023065) which allows for spin currents of the same orientation as discussed in both references have a similar underlying origin. Thus, given that there are several pathways for bulk spin currents to exist with spin direction along $\mathbf{m} \times (\mathbf{j} \times \mathbf{z})$ (with some higher-order contributions), it would seem best to refrain from singling out a particular bulk mechanism. The authors could certainly suggest that the “spin Hall precession effect” is a candidate but should inform the readers of other possibilities.

► **Our reply:**

We thank the reviewer’s comment on the mechanism of bulk contribution.

As the reviewer pointed out, there are possible effect from planar Hall effect to the present bulk contribution. Although, these mechanisms only show the presence of the spin current polarized along $\mathbf{m} \times (\mathbf{j}_e \times \mathbf{z})$ from the symmetry point of view and the detail mechanism of how it generates the spin current and how it diffuses remain unclear. Therefore, we focused on a spin-drift diffusion model-based mechanism (spin Hall precession effect) to show one of the explanations to reproduce our obtained layer thickness dependence. In the case of the magnetic spin Hall effect, the presence of the spin current polarized along $\mathbf{m} \times (\mathbf{j}_e \times \mathbf{z})$ was only discussed from the symmetry point of view.

Taking the reviewer’s comment into account, we discuss the possibility of multiple mechanisms other than spin Hall precession effect in the main text as follow. In addition to this, we refer Phys. Rev. Research 2, 023065 (2020) as reference 20 in the revised manuscript and added further explanation of the spin Hall precession effect in the supplementary information.

<Interfacial and bulk contribution to the charge-to-spin conversion, Page 4 line 35 ~ Page 5, line 5>

“Next, we discuss the bulk contribution. Multiple mechanisms have been proposed as an origin of the bulk contribution on σ_{MD} . For instance, Paulyac et al. proposed a spin swapping effect which generates an $\mathbf{m} \times \mathbf{y}$ polarized spin current within a diffusive ferromagnet [29]. However, this effect is considered to be weaker in FM with strong exchange interaction, such as Co/Ni and Co/Pt multilayers. The planar Hall effect [19] is another candidate that generate the $\mathbf{m} \times \mathbf{y}$ polarized spin current [19]. In addition to these effects, we propose an alternative mechanism to reproduce our layer thickness dependence of x_{MD} . Here, we focused on the mechanism originating from the combination of the spin Hall effect within FM and its interaction with local magnetization (Fig. 3c).”

<Discussion, Page 6 line 28 ~ 36>

“The other is that we experimentally show the first evidence of bulk contribution to the magnetic-dependent charge-to-spin conversion. Whereas the interfacial contribution is considered to originate from the spin-orbit precession effect, the origin of the bulk contribution is open to discussion because the magnetic-dependent spin current could arise from multiple mechanisms [20]. According to the first-principle calculation, planar Hall effect can generate a similar spin current with σ_{MD} symmetry [19]. Another possible source is the spin Hall precession effect as we proposed. However, it is difficult to quantify which mechanism plays a dominant role in bulk contribution from the present experiment. Further experimental and theoretical studies are necessary to classify the detail mechanisms of the spin-current generation.”

<Supplementary information S5, Page 13, line 7~14>

“We emphasize that multiple mechanisms can generate bulk spin current, as written in the main text. The spin Hall precession effect discussed below relates to the extrinsic mechanism of the spin Hall effect, such as skew scattering, where the spin transport could be diffusive due to the scattering and the spin polarization perpendicular to the local magnetization experiences dephasing [S7]. On the other hand, the spin current generated by the intrinsic mechanism is carried by electrons in perturbed eigenstates with same wave vector and does not experience the dephasing [S10]. The first principles calculations will be necessary to evaluate bulk spin current [S10,S11].”

Although the spin Hall precession effect is one plausible mechanism to explain the bulk contribution, it is worth noting that our manuscript show the first evidence of bulk contribution to the magnetization-dependent charge-to-spin conversion for the first time which would be one the significances in our work. Therefore, we added sentences to emphasized our significance “first experimental evidence of bulk contribution to the charge-to-spin conversion in ferromagnetic materials” in the revised manuscript as following.

<Abstract, Page 1 line14~16>

“Here, we achieved a large enhancement of magnetic-dependent charge-to-spin conversion by clarifying its mechanism. *Through thickness dependence, we revealed a coexistence of interfacial and bulk contributions to the magnetic-dependent charge-to-spin conversion.*”

<Introduction, Page2 line18~19>

“In this article, we report on the enhancement of spin current conductivity for magnetic-dependent charge-to-spin conversion in FM by clarifying its origin. *We show the first experimental evidence for the coexistence of interfacial and bulk contribution to the generation of σ_{MD} .*”

<Discussion, Page6 line28~29>

“We investigated the mechanism of charge-to-spin conversion originating from perpendicularly magnetized ferromagnetic material and its interface and ascertained two important features of magnetic-dependent charge-to-spin conversion. One is that the interfacial spin-orbit precession effect plays a dominant role in achieving high spin current conductivity. By engineering the interfacial band structure, we achieved up to three-fold enhancement in magnetic-dependent charge-to-spin conversion. The other is that we experimentally show the first evidence of bulk contribution to the magnetic-dependent charge-to-spin conversion.”

► **Referee #2 (Comments to author):**

The motivation of this paper is the control of out-of-plane magnetizations, yet in this work, the source layer has an out-of-plane magnetization and the free layer has an in-plane magnetization. The authors should at least comment on why this orientation was chosen. There are two cases of interest: 1) z -magnetization, x -electric field, z -spin flow, x -spin direction and 2) x -magnetization, x -electric field, z -spin flow, z -spin direction. Both cases are allowed by symmetry in ferromagnets. The first case is studied in this experiment whereas the latter is more useful for controlling magnetization dynamics in PML films. Certainly, both cases are related by the parameter σ_{MD} , but given high-order-in-magnetization contributions, the spin currents in each of these cases could be different. It would be useful for the authors to at least discuss how the measurements done in this geometry will help optimize the control of out-of-plane magnetizations in other geometries.

► **Our reply:**

We thank the reviewer for the suggestion for adding a discussion on the measurement geometry. In the present geometry, it can quantitatively evaluate the ζ_{MD} . This would be a great advantage over geometry 2) case (x -magnetization, x -electric field, z -spin flow, z -spin direction) in terms of material designing. Previous work has demonstrated the presence of the σ_{MD} in geometry 2) case using current induced coercive field shift and field free switching [Baek *et.al* Nat. Mater. **17**, 509 (2018)]. Although, this method contains various magnetic properties and complex of magnetization reversal process which make quantitative evaluation of ζ_{MD} to be difficult. ST-FMR measurement may be

possible in geometry 2) case using perpendicularly magnetized free layer as torque detector. However, accurate linewidth modulation becomes difficult in this case because the linewidth of perpendicularly magnetized system often broadens compared to in-plane magnetized system. Furthermore, it is technically difficult to fix the magnetization direction of the in-plane magnetized source layer resulting more complex layer structure (such as using exchange bias layer underneath the source layer). Regarding these facts, we believe that our device geometry is suitable to detect the spin current generation from ferromagnets.

Accordingly, we added extra paragraph in the supplementary information S4 as following.

<Supplementary Information S4, Page11 line20 ~ Page12 line4>

“The above formalism is applicable to analyze the spin-torque FMR spectrum in the other geometries by changing the variables such as the magnetic field and the spin-polarized directions. For example, a spin-torque FMR spectrum in an in-plane magnetized source layer and a perpendicularly magnetized free layer could be analyzed in a similar way. In such geometry, we can evaluate out-of-plane spin polarized spin current originate from the magnetic-dependent charge-to-spin conversion. We note, however, that the effective damping coefficient in perpendicularly magnetized layers are much broad compared to in-plane magnetized system. This makes the quantitative evaluation of the linewidth modulation difficult in this geometry. In addition, it is technically difficult to change the magnetization direction of the in-plane magnetized source layer, which makes it difficult to study the dependence of the spin polarization on the magnetization direction from a single device. Regarding these facts, we performed spin-torque FMR measurement in a perpendicularly magnetized source layer and an in-plane magnetized free layer to identify the existence and symmetry of the magnetic-dependent charge-to-spin conversion induced spin current.”

As the reviewer pointed out, σ_{MD} may be different due to high-order-in-magnetization contributions. But we note that the sign of σ_{MD} does not change with the geometry. This was confirmed by carrying out an additional SOT-induced magnetization switching experiment using geometry 2) case. As a result, we observed a SOT-induced chiral switching behavior and it is well explained by σ_{MD} obtained from ST-FMR measurement which is geometry 1) case. In the revised manuscript, we added this additional experiment in supplementary information section S8.

We hope our response to the reviewer’s comment and the additional experiment result would resolve the reviewer’s concerns.

► Referee #2 (Comments to author):

I believe this paper provides useful insight into the various spin current generation mechanisms in ferromagnetic trilayers.

► Our reply:

We thank the reviewer for his/her positive evaluation to our original manuscript.

► Referee #2 (Comments to author):

Overall, I consider the results and conclusions to be fairly convincing. However, it is worth raising an additional concern. Thickness-dependent measurements are one of the few ways to isolate interfacial contributions. However, since they require measurements to be performed over different samples, it is difficult to ensure that each sample has a similar interface, and that only the thickness of one layer is significantly different. However, given the lack of better approaches, it is worth performing these experiments. The results could be strengthened if more was said about the quality of the interfaces in each of the samples.

► Our reply:

We thank the reviewer for the comment on the quality of the interfaces in our system. To resolve the reviewer's concern, we collected a cross section TEM images of the series C samples. Figure R3 shows the HAADF-STEM image and the EDX elemental mapping of two series C samples with Ni concentration $x = 0\%$ and $x = 69\%$ interface. From the EDX mapping, we confirmed a relatively sharp interface between Cu layer and PML (Co/Ni multilayers). Because the $x = 0\%$ sample has a same interface with series A and B samples, the quality of the interface is good enough to ensure the similar interface between each sample with same interface. In addition to the quality of the interface, the Ni element in the $x = 69\%$ sample is concentrated near the interface compared to $x = 0\%$ sample indicating the interface is significantly engineered between two samples. This was also confirmed from the EDX line profile of each sample shown in Figure R4. These results ensure that controlling sub-nm-thick layer near the interface is an effective way to manipulate NM/FM interface.

In the revised manuscript, we added the above results to the supplementary information section S1. We hope our response to the reviewer's comment and the TEM analysis results would resolve the reviewer's concerns.

Figure R3: HAADF-STEM image and EDX elemental mapping: Series C sample with (a) $x = 0\%$ (Co interface) and (b) $x = 69\%$ ($\text{Co}_{31}\text{Ni}_{69}$ interface).

Figure R4: EDX line profiles of Series C samples: (upper panel) $x = 0\%$ (Co interface) and (lower panel) $x = 69\%$ ($\text{Co}_{31}\text{Ni}_{69}$ interface). The Ni concentration in the Co/Ni multilayer region is plotted by the open circles.

► Referee #3 (Comments to author):

In this manuscript, the authors demonstrate strong charge-to-spin conversion in ferromagnet. They utilize Co/Ni multilayers as the spin source, which shows bulk perpendicular magnetic anisotropy and benefits the thickness dependent analysis. This configuration also excludes the spin anomalous Hall effect as the origin. Both theoretical calculation and experimental results are concise, and the giant charge-to-spin conversion efficiency, which is an order of magnitude larger than the materials with reduced symmetry such as WTe₂, is exciting. However, the mechanism is still not clarified clearly enough as following, therefore I suggest this work to be further evaluated by Communications Physics.

► Our reply:

We thank the reviewer for his/her positive evaluation to our original manuscript. In the revised manuscript, we have improved the main text and supplementary information to clarify the mechanism of the spin orbit precession effect and the spin Hall precession effect. We hope our response to the reviewer's comment would resolve the reviewer's concerns. In addition to this, we have carried out additional experiments to strengthen our results such as spin pumping experiment and SOT-induced magnetization switching. Therefore, we would like the revised manuscript to be considered for publication in Nature Communications.

► Referee #3 (Comments to author):

In abstract, it is declared that "The interfacial contribution to charge-to-spin conversion is found to be dominant". However, this is only true when the Co/Ni multilayer is thin. The bulk contribution is also non-negligible from the thickness dependence result.

► Our reply:

We thank the reviewer's comment. As the reviewer pointed out, both the interfacial and bulk contribution to charge-to-spin conversion coexist in Co/Ni multilayer. Important point is that the interfacial contribution is always greater than the bulk contribution from our experimental result because the sign of the ξ_{MD} did not reverse from the thickness dependence result. Therefore, our statement "The interfacial contribution to charge-to-spin conversion is found to be dominant" does not mean that the bulk contribution is negligible but interfacial contribution plays an important role in the magnetization-dependent charge-to-spin conversion than bulk contribution.

► Referee #3 (Comments to author):

The detailed mechanism of bulk spin-Hall precession is not explained clearly. Why such spin would not undergo dephase? Why it would precess from one orthogonal-to-magnetization direction to the other?

► **Our reply:**

We thank the reviewer's useful comment to clarify the mechanism of the spin Hall precession effect.

In our model, the spin current generated by the bulk spin Hall effect undergoes the dephasing. The dephasing is described by the last term of Eq. (S5-5) in Supplementary Information. We note that the ferromagnetic thickness-dependence of the interfacial contributions to the spin conductivities reflect the dephasing process of spin current. This argument can be confirmed from, for example, Eq. (S5-17)-(S5-19), where $\Gamma_{x,y}$ characterize the amounts of the MD and MI spin currents generated at the interface. We note that the strength of the spin-transfer torque (or called damping-like torque) acting on the free layer is smaller than $\Gamma_{x,y}$ because a part of the spin current relaxes in PML due to dephasing and therefore does not contribute to the spin-transfer torque. The strength of the spin-transfer torque which really acting on the free layer is characterized by $\tilde{\Gamma}_{x,y}$, which includes the effect of the dephasing through the renormalization given by Eq. (S5-19). The parameters $\tilde{\Gamma}_{x,y}$ are proportional to the interface related spin Hall angles $\Theta_{x,y}$ as can be seen in Eqs. (S5-25) and (S5-26). Therefore, the dephasing of spin current is certainly taken into account in our analysis. Similarly, the thickness dependence of the bulk contribution to spin conductivity also reflects the spin dephasing, as written in the second paragraph in S5.5, Supplementary Information. The fact that both MI and MD spin conductivities (ξ_{MI} and ξ_{MD}) depend on the ferromagnetic thickness indicates the presence of the spin dephasing.

The spin current can, however, change its polarized direction because of the precession around the magnetization due to the exchange interaction before it completely relaxes by the dephasing. We understand that this is a controversial issue related to the mechanism of spin-transfer in nanoscale. The original work by Slonczewski [J. Magn. Magn. Mater. **159**, L1 (1996)] on the proposal of spin-transfer torque uses the ballistic spin-transport theory and assumes a fast dephasing near the ferromagnetic interface, where the length scale of the dephasing is on the order of angstrom. The first principle calculations also argue a similar conclusion; please see, for example, A. Brataas *et al.*, Phys. Rep. **427**, 157 (2006). If the dephasing length is too short, the spin polarization immediately becomes zero before changing the spin-polarized direction due to the precession. On the other hand, S. Zhang, and P. M. Levy, as well as their colleagues, had argued a relatively long dephasing length about 1~2 nm; please see, for example, S. Zhang *et al.*, Phys. Rev. Lett. **88**, 236601 (2002) and *ibid* **93**, 256602 (2004). When the dephasing length is relatively long as such, the spin polarization can change its polarized direction before the dephasing. This was also experimentally investigated by several groups which spin dephasing length can reach up to 1~3 nm; for instance, the following papers.

- T. Taniguchi *et al.*, Appl. Phys. Express **1**, 031302 (2008).
- X. Qiu *et al.*, Phys. Rev. Lett, **117**, 217206 (2016).

Several efforts have been made to clarify the validities of these theories; however, as far as we know, no solid conclusions are obtained yet. Some calculations support the Slonczewski's arguments, whereas some experiments support the Levy's arguments; see, for example, the following papers.

- M. Zwierzycki *et al.*, Phys. Rev. B **71**, 064420 (2005).
- M. D. Stiles and A. Zangwill, Phys. Rev. B **66**, 014407 (2002).
- W. Chen *et al.*, Phys. Rev. B **74**, 144408 (2006).
- T. Taniguchi *et al.*, Appl. Phys. Express **1**, 031302 (2008).
- T. Taniguchi *et al.*, Mod. Phys. Lett. B **22**, 2909 (2008).
- T. Taniguchi *et al.*, Phys. Rev. B **78**, 224421 (2008).
- A. Gosh *et al.*, Phys. Rev. Lett. **109**, 127202 (2012).

The fitting result, $\lambda_j = 1.75$ nm, in our study is close to the Levy's argument. We note, however, that the aim of the present work is not to investigate the dephasing length deeply.

In summary, the spin current undergoes the dephasing; therefore, the spin conductivity depends on the ferromagnetic thickness. Before the spin completely relaxes due to the dephasing, however, it can precess around the magnetization due to the exchange interaction and change its direction from one orthogonal-to-magnetization to the other.

In the revised manuscript, we added the comment on how the spin dephasing length affects the spin Hall precession effect as following.

<Interfacial and bulk contribution to the charge-to-spin conversion, Page 5 line 9~15>

“As a result, not only the ordinary spin current induced by the spin Hall effect (we define as spin Hall current with polarization $\sigma_{SHE} \propto \mathbf{j}_e \times \mathbf{z}$ but also an alternative spin current with an $\mathbf{m} \times \sigma_{SHE}$ spin orientation is generated. This spin current strongly depends on spin dephasing length λ_j . When λ_j is too short (a few Å), σ_{SHE} immediately vanishes before $\mathbf{m} \times \sigma_{SHE}$ spin orientation is generated. However, when λ_j is relatively long (1~2nm) [30], which has been experimentally demonstrated by several groups [32,33], $\mathbf{m} \times \sigma_{SHE}$ spin orientation can be generated before the dephasing. This alternative charge-to-spin conversion mechanism, which we call the spin-Hall precession effect, is different from the interfacial spin-orbit precession effect because the source is the spin Hall current in

the spin Hall precession effect whereas it is the spin polarized charge current in the spin orbit precession effect.”

► **Referee #3 (Comments to author):**

The spin-orbit precession is still dominant for ξ_{MD} in this system. I don't understand why they could realize such a large efficiency. Besides, this comparison is based on the materials with reduced symmetry, but the mechanisms are totally different. What if compare to the similar ferromagnetic trilayers.

► **Our reply:**

We thank the reviewer for his/her question. Although the reason for the larger torque efficiency is unclear at this point, this is one of the importance of our present work which can boost the further research. We should note that the first principle calculation result of ξ_{MD} in Co/Cu interface [Amin *et.al* Phys. Rev. Lett. **121**, 136805 (2018)] is in the same order with our ST-FMR results indicating the large efficiency does not come from the derivation error of the experiment.

So far, there is no way to compare ξ_{MD} with other ferromagnetic tri-layers because there are few reports on the quantitative estimation of spin current conductivity ξ_{MD} as far as we know. For example, Baek *et.al* Nat. Mater. **17**, 509 (2018) only quantitatively estimated ξ_{MI} but not ξ_{MD} . In their manuscript the presence of ξ_{MD} was demonstrated through coercive field shift and field-free switching experiments.

► **Referee #3 (Comments to author):**

In Fig. 3a, magnetic dependent spin current conductivity decreases with the increase of the PML thickness. But the theoretical fitting shows an opposite tendency when the PML thickness is very small. How do you explain this divergency between calculating and experiment? Since the interface and bulk have opposite contribution to the charge-to-spin conversion, is it possible to manipulate the sign of magnetic dependent spin current conductivity by further increasing the PML layer thickness?

► **Our reply:**

We thank the reviewer for providing an important comment to clarify the difference between the interface and bulk contributions. We want to mention that the non-monotonic behavior in the theoretical fitting is not the divergency between the experiment and the calculation. Here, we explain this in terms of interfacial and bulk contributions to the fitting.

Let us firstly focus on the interfacial contribution. The fitting results indicate that magnetic-independent (MI) interface contribution is larger than the magnetic-dependent (MD) contribution;

please see MI and MD-induced interface related spin Hall angles (Θ_x and Θ_y) in Table S5-1 of Supplementary Information S5. The results explain why the ζ_{MD} shows non-monotonic behavior in very thin limit. In the zero-thickness limit of the PML, the values of the ζ_{MI} and ζ_{MD} reflect the Θ_x and Θ_y , respectively. However, with slightly increasing the ferromagnetic thickness from zero, MI spin current, polarized to the “-y”-direction, is converted to the spin current polarized to the “+x”-direction due to the precession of spin polarization around the magnetization. This converted spin has the same polarity to MD spin current and dominates in the ζ_{MD} because the Θ_y is much larger than Θ_x . As a result, the (net) MD spin conductivity increases with increasing the ferromagnetic thickness. A further increase of the ferromagnetic thickness leads to an oscillating behavior of ζ_{MD} due to the precession. In addition, because of the fast dephasing of spin current, ζ_{MD} from the interface contribution saturates immediately. These are the reason why the interfacial MD spin conductivity shows non-monotonic behavior near the zero-thickness limit.

On the other hand, the bulk contribution changes more slowly with increasing the ferromagnetic thickness due to the following reason. The thickness dependence of the spin conductivity is determined by, roughly speaking, the difference between the generated and relaxed spin currents. In the case of the interfacial contribution mentioned above, the amount of the generated spin current is just determined by the interface effect and is independent of the ferromagnetic thickness. Therefore, the thickness dependence of the interfacial contribution solely depends on the amount of the relaxed spin current. On the other hand, in the case of the bulk contribution, the amounts of both generated and relaxed spin currents depend on the ferromagnetic thickness. Accordingly, the bulk contribution changes slowly with increasing the ferromagnetic thickness. This point was already written in S5.5 in Supplementary Information. As a result, the total spin conductivity almost overlaps with the interfacial contributions near the zero-thickness limit.

Regarding these considerations, the non-monotonic behavior of ζ_{MD} relates to the contribution from interfacial MI spin current, which rotates around the magnetization and is converted to MD contribution. The bulk contribution plays a minor role in the zero-thickness limit due to the slow dependence on the ferromagnetic thickness. Due to the fast dephasing, however, it is difficult to measure such a non-monotonic behavior experimentally. However, we emphasize that this fact does not mean the existence of divergency between the experiments and theory.

It may be possible to manipulate the sign of the magnetic dependent spin current conductivity by increasing the PML layer thickness. Unfortunately, we could not further increase the thickness of PML layer in the present system because PML starts to prefer multi-domain structure under in-plane field. The further thickness dependence will be our future work.

► **Referee #3 (Comments to author):**

The authors claim that both the Rashba effect and spin polarization contribute to the interface-related charge-to-spin conversion, but they did not explain why. If these two effects are linear with the Ni concentration and have opposite sign, why ξ_{MD} drop at Ni 100%?

► **Our reply:**

We thank the reviewer for his/her question. As the reviewer pointed out, both the interface Rashba effect and spin polarization play an important role in the interface-related charge-to-spin conversion ξ_{MD} (which is spin-orbit precession effect). The explanation of how these factors contribute to the conversion phenomena is mentioned in the section “Origin of the charge-to-spin conversion” in the original manuscript (“interfacial and bulk contribution to the charge-to-spin conversion” in the revised manuscript). By increasing the Ni concentration, the interface Rashba effect is enhanced due to the enhancement of the work function difference whereas the spin polarization near the interface decrease because the spin polarization in Ni is smaller than that of Co. Therefore, we consider that the reduction of the spin polarization causes the ξ_{MD} drop at Ni 100%.

► **Referee #3 (Comments to author):**

The authors should clarify that the solid lines in Fig. 4c are just guide to the eye.

► **Our reply:**

As the reviewer pointed out, we added the following caption in Fig. 4c: “The solid line is guide to the eye.”

Reviewer comments, second version:

REVIEWERS' COMMENTS:

Reviewer #1 (Remarks to the Author: Overall significance):

The authors answered my questions well. I am quite satisfied with their reply.

Reviewer #1 (Remarks to the Author: Strength of the claims):

I think this work is quite valuable to further research. Some previous work were provided, and this work has extended the research. I think this research may start some new research on spin torque generation by magnetic materials.

Reviewer #2 (Remarks to the Author: Overall significance):

I believe the authors have addressed my concerns from my initial review. I can recommend that this paper be published given one required change. described below.

The language the authors use to discuss bulk mechanisms originating from the planar Hall effect is misleading. Based on the current prose, some readers may assume that the spin current of interest is just a spin-polarized version of the planar Hall effect. However, this cannot be true since the spin orientation is orthogonal to the magnetization. Instead, the authors could say that the spin current is "related" to the planar Hall effect, in that the spin current and the planar Hall current share the same origin.

It may help to clarify (as far as I understand it) this possible bulk mechanism. To calculate the planar Hall effect, one may first assume that an applied electric field creates a nonequilibrium occupation of carriers. The nonequilibrium occupation can be obtained, for example, using the relaxation time approximation. Then, one may calculate the perpendicularly flowing charge current given the nonequilibrium occupation from the applied electric field. This perpendicularly-flowing charge current will have the magnetization dependence expected of the planar Hall effect. Thus, the calculated charge current is one contribution of the planar Hall effect. If one then calculates the perpendicularly-flowing spin currents under these same conditions, one of the allowed spin currents should have the orientation of interest (i.e. just like the spin-orbit precession current). In this sense, the spin currents of interest generated in ferromagnetic materials are "related in origin" to the planar Hall effect.

While it is not necessary to provide such detail in the manuscript (pointing to the appropriate references should suffice), it is important to clarify that the spin currents of interest originating in ferromagnets can arise from spin swapping and from a spin current closely related to the planar Hall effect.

Given that this clarification is added in some form, I can recommend publication.

Reviewer #3 (Remarks to the Author)

see attachment

The authors performed careful experiments on the SOT in the PML/NM/IML trilayer. Although the trilayer system has been investigated by several groups, the present work provided insight on the origin of torques and observed a strong conversion, which should be quite interesting and useful for the spintronics community, especially the SOT field. The authors carefully addressed the comments and questions from the previous three referees. The quality of the work has been greatly improved with new data (especially the ST-FMR measurements and STEM characterizations). The authors carried out systematic measurements. For example, they vary different layer thickness and composition. This kind of experiments supports the reproducibility. They also give reasonable error bar in the data. The work is ready for publication, I would say, up to the standard of Nat. Commun. But there are two tiny issues should be further addressed.

1. The main concern of the present work is the spin current from the PML layer to IML layer. Whether any opposite effect occurs in the present system, e.g., from the IML to PML layer? Whether these two torques and corresponding switching can be obtained. Some relevant discussions are suggested to be added.
2. For Fig. S9(d) and (e), it is easy to understand their anticlockwise and clockwise. But why the shape of the switching curves is also different?
3. Out-of-plane polarized spin current and the corresponding field-free SOT switching was observed recently in antiferromagnet/ferromagnet (both PML and IML) system, Nat. Mater. 20, 800 (2021), which should be referred, since it is a quite related work.

Author rebuttal, second version:

Replies to Reviewers' Comments

GUIDEDOA-21-00047A

Giant charge-to-spin conversion in ferromagnet via spin-orbit coupling

By Y. Hibino, T. Taniguchi, K. Yakushiji, A. Fukushima, H. Kubota, and S. Yuasa

We would like to thank all the referees for the careful reading of our manuscript and for his/her valuable comments that helped us to improve the manuscript. Here, our responses to the referees' comments (shown as blue characters) are shown. The revised parts are marked with yellow in the main text.

► Referee #1 (Comments to author):

The authors answered my questions well, I am quite satisfied with their reply.

I think this work is quite valuable to further research. Some previous works were provided, and this work has extended the research. I think this research may start some new research on spin torque generation by magnetic materials.

► Our reply:

We are very pleased to hear that the referee had satisfied with our replies. We also want to thank the referee for being supportive to the publication of our work.

► Referee #2 (Comments to author):

I believe the authors have addressed my concerns from my initial review. I can recommend that this paper be published given one required change described below.

The language the authors use to discuss bulk mechanisms originating from the planar Hall effect is misleading. Based on the current prose, some readers may assume that the spin current of interest is just a spin-polarized version of the planar Hall effect. However, this cannot be true since the spin orientation is orthogonal to the magnetization. Instead, the authors could say that the spin current is "related" to the planar Hall effect, in that the spin current and the planar Hall current share the same origin.

It may help to clarify (as far as I understand it) this possible mechanism. To calculate the planar Hall effect, one may first assume that an applied electric field creates a

nonequilibrium occupation of carriers. The nonequilibrium occupation can be obtained, for example, using the relaxation time approximation. Then, one may calculate the perpendicularly flowing charge current given the nonequilibrium occupation from the applied electric field. This perpendicularly-flowing charge current will have the magnetization dependence expected of the planar Hall effect. Thus, the calculated charge current is one contribution of the planar Hall effect. If one then calculates the perpendicularly-flowing spin currents under these same conditions, one of the allowed spin currents should have the orientation of interest (i.e. just like the spin-orbit precession current). In this sense, the spin currents of interest generated in ferromagnetic materials are “related in origin” to the planar Hall effect.

While it is not necessary to provide such detail in the manuscript (pointing to the appropriate references should suffice), it is important to clarify that the spin currents of the interest originating in ferromagnets can arise from spin swapping and from a spin current closely related to the planar Hall effect.

Given that this clarification is added in some form, I can recommend publication.

► **Our reply:**

We thank the referee for the comment and for providing further explanation of the bulk mechanism in spin current generation. In the revised manuscript, we modified the explanation of the spin currents generated from ferromagnet to be an extrinsic mechanism, which are related in origin to the planar Hall effect. Accordingly, we revised the corresponding sentence as following:

<Interfacial and bulk contribution to charge-to-spin conversion, Page 5 line 5 ~ 6>

“Another candidate that generates the $\mathbf{m} \times \mathbf{y}$ polarized spin current is the extrinsic mechanism, which is related in origin to the planar Hall effect [20].”

► **Referee #3 (Comments to author):**

The authors performed careful experiments on the SOT in the PML/NM/IML trilayer. Although the tri-layer system has been investigated by several groups, the present work provided insight on the origin of torques and observed a strong conversion, which should be quite interesting and useful for the spintronics community, especially the SOT field. The authors carefully addressed the comments and questions from the previous three referees. The quality of the work has been greatly improved with new data

(especially the ST-FMR measurements and STEM characterizations). The authors carried out systematic measurements. For example, they vary different layer thickness and composition. This kind of experiments supports the reproducibility. They also give reasonable error bar in the data. The work is ready for publication, I would say, up to the standard of Nat. Commun.

► **Our reply:**

We are pleased to learn that the referee pointed out the novelty and importance of our work and support the publication of our manuscript. Moreover, we thank the referee for summarizing the critical points of our work.

► **Referee #3 (Comments to author):**

But there are two tiny issues should be further addressed.

1. The main concerning of the present work is the spin current from the PML layer to IML layer. Whether any opposite effect occurs in the present system, e.g., from the IML to PML layer? Whether these two torques and corresponding switching can be obtained. Some relevant discussions are suggested to be added.

► **Our reply:**

We thank the referee for the comment. In principle, the spin current from the IML can be generated and injected to PML layer as the referee pointed out. However, we think that this effect in the system shown in the main text is negligibly small to induce switching behavior, due to the following two reasons. One is that the shunting current ratio in IML is very low compared to PML, which is caused by high resistivity of Fe-B layer. Because of this, spin current generated from IML is very small compared to that generated from PML. The second reason is the high perpendicular magnetic anisotropy of the PML. This causes the magnetization dynamics such as ferromagnetic resonance and switching to be difficult in the present system. To check this, we have tried to observe the FMR of PML in the present system by applying out-of-plane magnetic field. However, we did not observe any significant FMR signal from PML. To summarize, the PML used in the main text is basically unaffected by the spin current from IML and can be regarded as a fixed spin source layer. Accordingly, we added the following sentence to discuss this.

<Spin current generation from ferromagnet, Page 3 line 26 ~ 29>

“We note that the IML (Fe-B layer) may also generate spin currents, resulting a torque

acting on PML. However, this torque is negligible in our system. The reason for this is that the spin current generated from IML is relatively small compared to PML due to high resistivity of Fe-B layer. In addition to this, high perpendicular magnetic anisotropy of PML makes it difficult to induce magnetization dynamics in PML.”

► **Referee #3 (Comments to author):**

2. For Fig. S9(d) and (e), it is easy to understand their anticlockwise and clockwise. But why the shape of the switching curves is also different?

► **Our reply:**

We thank the referee for the comment. The main reason of the shape difference between two magnetic configurations is that the applied maximum current is higher in the case of Fig. S9(d) than that of Fig. 9(e). As mentioned in the supplementary note 10, perpendicularly magnetized free layer used in the experiment can be easily demagnetized by the Joule heating. Therefore, we infer that the magnetization of free layer is fully demagnetized above current density of $1.5 \times 10^7 \text{ Acm}^{-2}$, which can be seen in Fig. S9(d). In the case of Fig.S9(e), we did not observe such demagnetized behavior because we did not apply switching current that much. We note that the threshold switching current density for both cases were confirmed to be $\approx 1.2 \times 10^7 \text{ Acm}^{-2}$.

► **Referee #3 (Comments to author):**

3. Out-of-plane polarized spin current and the corresponding field-free SOT switching was observed recently in antiferromagnet/ferromagnet (both PML and IML) system, Nat. Mater. 20, 800 (2021), which should be referred since it is a quite related work.

► **Our reply:**

We thank the referee for pointing out the related work. We have added the corresponding work as reference 14 in the revised manuscript. In addition, we added following comment in the main text.

<Introduction, Page 1 line 37 ~ Page 2 line 2>

“To overcome this geometrical restriction, symmetry-reduced materials, such as transition-metal dichalcogenides [11] and non-collinear antiferromagnets [12,13], have been proposed as candidate materials whose unique crystal structure and spin texture break the in-plane symmetry. Some of them have demonstrated a field-free switching

behavior [14].”